# Thermal Effect on Heat Shock Protein 70 Family to Prevent Atherosclerotic Cardiovascular Disease

**DOI:** 10.3390/biom13050867

**Published:** 2023-05-20

**Authors:** Masayo Nagai, Hidesuke Kaji

**Affiliations:** 1Central Research Facility, Aino University, Osaka 567-0012, Japan; 2Division of Physiology and Metabolism, University of Hyogo, Kobe 651-2197, Japan

**Keywords:** HSP70, heat stress, ASCVD, skeletal muscle

## Abstract

Heat shock protein 70 (HSP70) is a chaperone protein induced by various stresses on cells and is involved in various disease mechanisms. In recent years, the expression of HSP70 in skeletal muscle has attracted attention for its use as a prevention of atherosclerotic cardiovascular disease (ASCVD) and as a disease marker. We have previously reported the effect of thermal stimulation targeted to skeletal muscles and skeletal muscle-derived cells. In this article, we reported review articles including our research results. HSP70 contributes to the improvement of insulin resistance as well as chronic inflammation which are underlying pathologies of type 2 diabetes, obesity, and atherosclerosis. Thus, induction of HSP70 expression by external stimulation such as heat and exercise may be useful for ASCVD prevention. It may be possible to induce HSP70 by thermal stimulus in those who have difficulty in exercise because of obesity or locomotive syndrome. It requires further investigation to determine whether monitoring serum HSP70 concentration is useful for ASCVD prevention.

## 1. Introduction

### 1.1. HSP Family

Heat shock proteins (HSPs) are widely known as stress response proteins. HSP family consists of HSP27, HSP40, HSP60, HSP70, HSP90, HSP110, and HSP170. Human HSP70 are a group of 13 members encoded by HSPA genes: HSPA1A, HSPA1B, HSPA2, HSPA4, HSPA5, HSPA6, HSPA7, HSPA8, HSPA9, HSPA12A, HSPA12B, HSPA13, and HSPA1.4 [1,2].

### 1.2. Molecular Mechanism of HSP70 Induction and Release

Expression of HSP is induced by various stressors including oxidative stress [3], heat stress [4], ischemia [5], exercise [6], and metabolic stress [1].

Factors such as heat shock factor 1(HSF1) [7,8], heat shock protein binding protein 1(HspBP1) [9], and HSP70-ATPase [10] regulate HSP70 activity. HSF1 is a transcription factor of HSP70 by binding to a specific promoter sequence called HSE (Heat shock element) [7,11]. The human genome encodes six HSF proteins, which normally reside in the cytoplasm as inactive monomers and translocate to the nucleus upon activation [7].

The carboxy terminus of Hsp70-binding protein (CHIP) has been reported [8] as a factor that modulates HSP70 induction. CHIP is a cytoplasmic protein (ubiquitin ligase) whose amino acid sequence is highly conserved across species and is most abundantly expressed in cardiac muscle, skeletal muscle, and brain [10]. CHIP enhances HSF1-mediated HSP70 induction during acute stress and is also involved in the regulation of HSP70 expression through a non-HSF1-mediated pathway [8]. 

Although the biological significance of HspBP1 is poorly understood, it belongs to a family of co-chaperones that regulate HSP70 activity [12]. It has been reported that the expression level of HspBP1 is markedly increased following oxidative stress [9]. HSP70 exerts a chaperone function by ATPase activation. HspBP1 is an inhibitor of CHIP, but it is still unknown whether HspBP1 cooperates with CHIP in regulating the heat shock response [9] (Figure 1).

Upon cellular stress, the intracellular HSP70 (iHSP70), which initially resides within the cell, is translocated into the circulation via such mechanisms as cell necrosis or ATP-binding cassette (ABC) transporters [13,14]. This HSP was named as extracellular HSP70 (eHSP70). iHSP70 exerts an anti-inflammatory action by the heat shock response. In contrast, eHSP70 may mediate proinflammatory pathways [15] and associate with insulin resistance in type 2 diabetes mellitus (T2DM). So, it is proposed to exploit the ratio of eHSP70 to iHSP70 as a biomarker [15]. 

### 1.3. Thermal Stimulation and HSP70 Family

Accumulating evidence suggests the role of thermal stimulation in preventing athero sclerotic cardiovascular disease (ASCVD). We have previously reported the effect of thermal stimulation on human skeletal muscles in vivo and skeletal muscle-derived cells (SMDC) in vitro. HSP70/HSP72 was measured before and after thermal load in our previous study [16]. Thus, this review focused on the role of the thermal effect on serum and muscle HSP70. The HSP70 family also functions as a chaperone and is involved in a number of physiological mechanisms [7]. Thus, it has gained attention as a target for disease prevention and a possible biomarker of disease [7,17]. Inducible proteins such as HSPs may be important for mitigating lifestyle risk factors and disease management [18].

HSP72, a family of HSP70 encoded by the HSPA1A gene, is also known as a heat-responsive protein, as HSP72 is overexpressed with thermal stimulation. Weekly in vivo hyperthermia improved glucose tolerance, elevated muscle strength and hepatic HSP72 protein content, and reduced muscle triglyceride storage [19]. 

Thermal stimulation of skeletal muscle may be a way to promote glucose metabolism through HSP activity [20]. HSPs have been shown to reduce oxidative stress, inhibit inflammatory pathways, and enhance the metabolic properties of skeletal muscle [21]. Conversely, it has been reported that insulin resistance may be induced by chronic inflammation under an imbalanced HSP70 status [22].

Previously, we have examined whether thermal stimulation of SMDC is useful to prevent ASCVD [23]. In the present article, we discuss our findings as summarized in Table 1 and review the literature demonstrating the effect of HSP70 expression in the skeletal muscle on ASCVD.

HSP70 is one of the most induced molecular chaperones under stress [8]. HSP72 are in the nucleus and cytoplasm [25]. HSPs constitute 5–10% of the total protein content of cells under physiological conditions and can increase by up to 15% under stress [26].

## 2. Thermal Effects through HSP70 on the Underlying Pathology of ASCVD 

Table 2 summarizes previous reports on how HSP70 is involved in the pathogenesis of ASCVD.

### 2.1. Insulin Resistance and Chronic Inflammation

Insulin resistance precedes important pathologies such as cardiovascular atherosclerosis. Both systemic and local heat stimulation activate the HSP72 heat shock pathway and induce the expression of related genes. This, in turn, leads to weight loss and improved glucose metabolism. It has been reported that the expression of HSPA1A/HSPA1B (HSP70/HSP 72) was decreased by obesity, insulin resistance [19], and ageing [27]. It has been shown that the HSP70 family has preventative effects against a high-fat diet, (HFD)-induced obesity, and insulin resistance [19,39]. Multiple review articles [25,40] also agree that HSP72 levels decline as a metabolic syndrome with insulin resistance progresses to T2DM, and HSP72 expression levels are closely associated with obesity.

HSP72 inhibits the activation of stress kinases such as c-Jun-NH2-terminal kinase (JNK), which is known to interfere with the insulin signaling pathway. JNK promotes insulin resistance by inhibiting the phosphorylation of insulin receptor substrate 1 (IRS-1), a key protein in the insulin signaling cascade [41]. As diabetes progresses, JNK becomes more activated while HSP72 expression decreases [25]. The induction of HSP72 directly inhibits JNK activation, and this has been indicated to improve insulin sensitivity and glucose tolerance at both skeletal muscle and systemic levels. Furthermore, in vivo heat treatment was shown to reduce JNK activation in the skeletal muscle of old rats and HFD-feeding rats [28,29]. HSP72 induction by chronic heat treatment was also shown to protect skeletal muscles against obesity-associated insulin resistance [29]. Not only chronic but also acute heat treatment may be effective to improve insulin resistance. Gupte et al. [29] exposed the lower body of old rats to a short period of thermal stimulation using a thermal blanket and demonstrated that the heat exposure improved insulin-stimulated glucose intake in the skeletal muscle. The authors further suggest that the underlying mechanism of this effect was the inhibition of JNK via HSP72. Taken together, these studies suggest that HSP72 plays a critical role in improving insulin sensitivity and glucose tolerance that results from reduced JNK activation. Thus, interventions targeting HSP72 may be effective to prevent ASCVD, and both chronic and acute thermal stimulation may be effective to increase the expression of HSP72. A correlation among insulin resistance, T2DM, and HSP70 is hopeful for T2DM management, and the target factor HSP70 may be modulated through interventions such as long-term physical exercise or hot tub therapy [25].

It is widely accepted that the level of HSP70 expression increases under stressful conditions. HSP72 was also shown to prevent age-related [27] and HFD-induced obesity and insulin resistance [39]. This observation may be attributed to a number of mechanisms, including blockage of JNK signaling by HSP72 in skeletal muscle, activation of AMPK and the prevention of intramuscular lipid accumulation, and enhanced mitochondrial generation and oxidative metabolism via increased sirtuin activity [39]. JNK is a key modulator of intracellular signaling, and its activation in skeletal muscle is one of the factors that lead to insulin resistance. In fact, a study demonstrated that HFD-induced JNK activity was suppressed in mice with HSP72 overexpression [39]. 

Skeletal muscles play an important role in insulin-stimulated glucose uptake [42] as shown by several studies examining changes in HSP expression in skeletal muscles. We have previously examined the change in gene expression after thermal stimulation in SMDC [23,43]. We have previously reported that thermal stimulation of SMDC changed the expression of genes involved in improving insulin resistance by promoting glucose uptake through various insulin-dependent and independent mechanisms [23]. We observed a significant increase in the level of HSP72 (HSP A1A) mRNA expression. Furthermore, increased HSP72 expression in skeletal muscle was associated with an increase in the expression of lipoprotein lipase (LPL), demonstrating that increased mitochondria in skeletal muscle leads to the suppression of insulin resistance [39]. Improvement in the insulin response without dietary restriction in mice with HSP72 overexpression further led to increased insulin-stimulated glucose uptake in the anterior tibia muscle. This effect was also observed in white and brown adipose tissues that do not overexpress HSP72, suggesting that the increased HSP72 expression within skeletal muscle also indirectly affects adipocytes [39]. Taken together, these findings indicate that thermal stimulation to skeletal muscle, a central regulator of metabolism, may be used effectively to have an impact on the systemic level.

On the other hand, it has also been reported that HSP70 promotes insulin resistance in tissues other than skeletal muscle [30]. Previous reports [30] have shown that HSP70 and GRP78 (Glucose-Regulated Protein 78 kDa) are secreted from the human jejunum, and a high-calorie diet increases circulating levels of HSP70 and insulin resistance. It has been reported that plasma HSP70 concentration is negatively correlated with insulin sensitivity. In vitro validation in the same study also indicated that HSPs inhibited Akt phosphorylation and glucose uptake in HepG2. The secretion of stress response proteins HSP and GRP from the small intestine is associated with insulin resistance as key mediators of intercellular signaling in response to nutrients [30]. In addition, it was considered that HSP70 and GRP78 bind to Toll-like receptors (TLRs) that stimulate adipogenesis and intracellular fat accumulation with the promotion of insulin resistance [30].

HSP72 is known to have an anti-inflammatory property. HSP72 may elicit an additional anti-inflammatory effect via localizing in the extracellular space and macrophages. In mammals, HSP72 binds with high affinity to macrophages and peripheral blood mononuclear cells and is taken up by endocytosis to localize within cells [44]. In the intra- and extra-cellular spaces, HSP72 inhibits the release of inflammatory cytokines [45]. By locally reducing inflammation, HSP72 may play an important role in reducing the risk of developing insulin resistance [46]. Inflammation is an event induced by JNK activity and may be mediated by HSP70. The role of iHSP70 is known to inhibit JNK activation through several mechanisms including direct protein contact to JNK [25]. Since chronic inflammation is an underlying condition of ASCVD, HSP72 may be effective in the prevention and treatment of ASCVD.

On the other hand, eHSP70 can activate NFκB and activator protein 1, thus stimulating the release of inflammatory cytokines and the generation of reactive oxygen species (ROS) [17]. 

Thus, eHSP plays an important role in the induction of cell-mediated immune responses. However, the mechanisms underlying the association between HSP70, and inflammatory diseases have not been fully elucidated.

Furthermore, there is a recent report on the ratio of iHSP70 to eHSP70 and inflammation [25]. Because iHSP70 has antagonistic roles in inflammation, it has been hypothesized to use the ratio, R = [eHSP70]/[iHSP70], as a marker of inflammation [25]. The previous study [25] validated the ratio of plasma eHSP70 to lymphocyte iHSP70. When no stimulation controls were set at R = 1, R values >5 indicated a proinflammatory state, while R values <1 indicated an anti-inflammatory state. 

### 2.2. Apoptosis

It is well known that HSPs play an important role in the apoptosis pathways. HSPs have been shown to interact with different key apoptotic proteins. As a result, HSPs can block essentially all apoptotic pathways, mostly involving the activation of cysteine proteases called caspases [46]. In our previous study [43], we have also demonstrated that increased HSP expression was associated with an increased level of nucleolar protein 3 (apoptosis repressor with CARD domain: NOL3), which codes for initiator caspases in the apoptotic pathway.

Overexpression of Hsp70 is known to block apoptosis at various stages. HSP70 inhibits the apoptotic pathway by acting as a chaperone, and overexpression of HSP70 was shown to suppress apoptosis [31]. Increased expression of HSP72 by thermal stimulation has the effect of suppressing muscle atrophy and promoting protein synthesis and muscle hypertrophy [35,36]. 

However, the relationship between the induction of HSP70 expression and the acquisition of resistance of cells to heat has not been clarified [47]. In our previous study [43], neither cell morphological change nor cell detachment was observed 20 h after the start of thermal stimulation. The number of cells seeded at the start of the subculture was comparable. Notably, thermal stimulation of SMDC not only led to a significant increase in the expression of HSP70, but it changed the expression of several apoptosis-related genes [43].

JNK expression is suppressed as HSP72 expression increases; it has also been shown that JNK inhibits mitochondrial respiration and increases the production of ROS to cause apoptosis [41]. JNK activation is an important step in the cellular response to stress; however, prolonged JNK activation can lead to cell damage and cell death [32]. An increase in HSP72 expression due to thermal stimulation may, in turn, reduce the activity of JNK. Consequently, it is unlikely that the thermal stimulation of SMDC resulted in JNK promoting apoptosis and causing the observed change in apoptosis-related gene expression [43]. Taken together, these findings suggested that thermal stimulation of SMDC led to some level of heat stress induced by apoptosis-related mechanisms and subsequently led to the activation of the repair mechanism via HSP70. 

Protein p53 is one of the important regulators of apoptosis. Demyanenko SV [33] reported that eHSP70 reduced the level of neuronal p53 protein and inhibited the pro-apoptotic effect of WR1065. This indicates the involvement of eHSP70 in regulating p53 activity through JNK-dependent signaling pathways [33]. Demyanenko. SV [34] also reported that eHSP70 reduced apoptosis in glial cells. iHSP70 has also been reported to inhibit apoptosis by inhibiting NFκB activation, nitric oxide production, and superoxide dismutase activity [17]. In recent years, pathways of apoptosis inhibition by iHSP70 and eHSP70 have been elucidated. Endogenous effects are the main factor when utilizing thermal stimulation effects in living organisms. However, when utilizing the effect of HSP70, it is necessary to separately verify the effects of iHSP70 and eHSP70.

### 2.3. Chaperone Functions and Skeletal Muscle

HSPs are also involved in the unfolded protein response (UPR) [48]. We previously demonstrated [43] that there was a reduction in the expression of UPR stress markers including activation of transcription factor 6 (ATF6) and HSP70 (HSPA5). This indicated that thermal stimulation elevated the chaperone functions of HSPs, leading to the activation of enzymes that are involved in protein folding. Thermal stimulation, therefore, may have reduced endoplasmic reticulum (ER) stress within cells. 

The chaperone function of HSP70 is also exerted in the recovery processes of post-exercise damaged tissue including reduction of inflammation and oxidative stress [38]. It is believed that this can be used as a biomarker to verify the preventive effects of exercise on various diseases [38]. Furthermore, thermal stimulation can enhance the chaperone function of HSP70. Bathing in chloride hot springs enhances HSP70-induced protein repair function by improving immunity such as NK cell activity immediately after bathing [37]. When skeletal muscle is exposed to heat stress, HSP70 is increased and exerts chaperone function to suppress muscle atrophy [35]. Accumulation of protein misfolds is involved in skeletal muscle atrophy. Hyperthermia significantly decreases muscle atrophy markers such as CD68, KLF, and MAFbx, and significantly increases muscle hypertrophy markers such as AKT, mTOR, and HSP70 [36]. These thermal stimulation therapies are thought to be effective for muscle maintenance in diabetic patients [36].

On the other hand, there is a report that HSP70 expression promotes muscle atrophy. eHSP70 induces muscle wasting in cancer patients [49]. In zebrafish, HSP70 is one of the factors when excessive exercise causes muscle atrophy [50]. In addition, both the pro-inflammatory and anti-inflammatory effects of HSP70 in rheumatoid arthritis have been reported, and further verification is required [51]. Therefore, the use of thermal stimulation effects should be discussed when muscle atrophy and rheumatoid arthritis are present.

## 3. Thermal Effects on Serum HSP70 Levels

In a resting state, the serum HSP72 concentration level does not show an apparent endogenous circadian rhythm and does not fluctuate in response to climate changes [52]. Therefore, changes in the serum HSP70 level due to thermal stimulation may reflect in vivo reactions, although the origin of HSPs in the blood is unknown [52]. A previous study [53] found significant differences in HSP70 levels, serum insulin levels, HOMA-IR, and superoxide dismutase levels between lean and overweight young men. There was a negative correlation between the concentration of HSP70 and insulin levels and HOMA-IR, which was associated with increased activity of antioxidant enzymes. Lubkowska. A [53] suggested that the extracellular concentration of HSP70 could be an important indicator of impaired glucose homeostasis. Recent studies also have shown that HSP72 is involved in various diseases including, but not limited to, those leading to ASCVD such as diabetes and atherosclerosis. Thus, HSP72 may be available as a disease biomarker. 

It has also been reported that after bathing, there was a change in the serum HSP70 concentration from 15 min to 2 days [37] and that there was a change in the serum HSP70 concentration immediately after 60 min of exercise [53]. The serum high-sensitivity HSP70 measured in our previous study is affected by several pathophysiological processes including recovery from exercise, the process of repairing damaged tissue, and the reduction of oxidative stress due to inflammation [38]. 

In our previous study, we applied thermal stimulation to the body using four different types of heating devices; however, we did not observe an increase in the serum HSP70 concentration [16]. While thermal stimulation of cells would certainly induce HSP72, the mild thermal stimulation we used in our study was likely insufficient to cause an increase in the serum HSP70 level. 

In other words, localized thermal stimulation of skeletal muscle may not be sufficient to cause an increase in the serum HSP70 concentration. However, we cannot exclude the possibility that thermal stimulation of skeletal muscles does not affect the mRNA expression of HSP70. Although HSP70 was previously thought to be an intracellular factor, recent evidence [13] demonstrates that it is also released into the extracellular space as stated above. Based on these findings, measurement of serum HSP70 may be an index of effectiveness or safety when we apply heat stimulation for the long term to prevent ASCVD.

After tissue stimulation, there will be some time from HSP70 mRNA expression to protein synthesis. Consequently, an increase in HSP70 protein in tissues may not be reflected in blood circulation. To date, there are no studies that examined the level of HSP70 in various tissues in comparison with the serum concentration. 

In our previous study [16], plasma HSP70 was decreased by heat stimulation using a hot towel, and no significant change in plasma HSP70 was confirmed by red bean bags, hot packs, and heating sheets. However, the skin temperature was significantly elevated. As the increase in serum HSP70 may reflect the leak from damaged cells or tissues, it was considered that no change or reduction of serum HSP70 levels reflects cell safety. 

There are few reports on baseline resting eHSP70. Future research should aim to thoroughly characterize individual lifestyle factors, physical activity levels, and anthropometric and physiological characteristics [54]. 

Moreover, to obtain a thermal stimulating effect on the living body, the effect of bathing is expected. In a report by Maeda [37] on spa therapy, the thermal effects of chloride springs are effective in raising body temperature, keeping warm, and promoting blood flow in a single bath. It is recognized for its psychological relaxation effect. It has been stated that bathing twice a day is more effective than bathing once a day, and it may be easier to adopt as an ASCVD risk-reduction activity.

## 4. Thermal Effects through HSP70 on Underlying Diseases of ASCVD

Table 3 summarizes the previous reports on the effects of HSP70 on cardiovascular dis-ease.

### 4.1. Type 2 Diabetes

Decreased iHSP70 and increased eHSP70 expression are found in patients with obesity and metabolic diseases including T2DM [25]. Circulating HSP70 levels in gestational diabetes correlate well with levels of glucose metabolism-related protein [53]. The underlying theory for elevated serum HSP70 in diabetic patients is that diabetes and associated oxidative stress may induce heat shock responses [55]. Although the exact in vivo mechanism of diabetes on serum HSP70 levels is still unclear, inflammation from advanced diabetic conditions increases HSP70 in non-insulin-sensitive diabetic tissues such as the endothelium [55]. It has also been suggested that serum HSP70 may originate from vascular endothelium, a tissue that is not particularly sensitive to insulin activity [55].

Hooper [57] also describes cell viability, particularly in tissues where insulin signaling occurs but does not result in glucose uptakes such as pancreatic β cells and neurons. Insulin resistance in these tissues may result in a low HSP state that leads to increased intracellular protein aggregation, further increasing tissue fragility. The loss of pancreatic β cells due to low HSP conditions and the resulting intracellular amyloid accumulation leads to further loss of insulin signaling [57]. In such a vicious cycle, HSP depletion in diabetes can make tissues more susceptible to stress, leading to increased diabetes-related mortality and organ failure. 

Furthermore, HSP72 contributes to sustaining increased insulin secretion in the presence of insulin resistance by making a direct protective contribution against β cell apoptosis [41]. Therefore, induction in the liver as well as skeletal muscle may be important for HSP70 strategies toward T2DM [41].

As mentioned above, HSP70 is well known to be released from immune cells under stress conditions. A comparison of patients with endometrial cancer confirms elevated plasma eHSP70 in the presence of diabetes [56]. As atherosclerosis progresses, the HSP70 concentration in plasma also increases [17]. It has been reported that the serum HSP70 concentration significantly increases as the duration of diabetes prolongs [55]. This observation is likely attributed to the fact that the patient is exposed to prolonged oxidative stress. On the other hand, it is known that iHSP70 is decreased and eHSP70 is increased in T2DM and obese individuals [25]. Expression of iHSP and HSF1 is also decreased in the skeletal muscle of diabetic and obese individuals, while eHSP is increased [62].

### 4.2. Atherosclerosis

There are various reports on the effects of HSP70 on atherosclerosis. While overexpression of HSP27 is protective and overexpression of HSP60 is atherogenic, the effect of HSP70 is considered inconclusive [7]. Future verification is an important issue.

Hooper et al. [57] state that raising HSPs reduces inflammation and improves insulin action. It is pointed out that chronic inflammation, the underlying pathology of ASCVD, is affected by a vicious cycle of metabolism. Obesity, a sedentary lifestyle, and a high-fat calorie diet accelerate this cycle by reducing insulin signaling, increasing inflammatory cytokines, and inducing a low HSP state [57]. Similarly, prolonged stress elevates stress hormones (cortisol, catechol amines, and glucagon), ultimately impairing insulin signaling and lowering HSP levels. Thus, HSPs decline is associated with insulin resistance, and inflammation [57]. Diet and exercise are known to raise HSPs, reduce inflammation, improve insulin signaling, and reverse the cycle leading to T2DM [57]. Thermal stimulation and exercise increase HSP70 and are used to improve insulin resistance in T2DM [19]. It can be considered that the increase in HSP70 concentration in chronic inflammation is partly due to the exertion of the chaperone function to maintain homeostasis against stress. Therefore, it is likely that an increase in HSP70 concentration does not promote atherosclerosis but is the result of atherosclerosis progression.

An elevated serum HSP72 concentration is associated with an increased number of vascular lesions [58]. HSPs act as chaperones, and the expression of HSP72 was shown to increase during myocardial infarction; this has an anti-apoptotic effect as it protects cardiomyocytes from undergoing apoptosis that results from ischemic injury [59]. A study also demonstrated that there was a positive correlation between the serum HSP70 level and systolic blood pressure in newly diagnosed hypertensive patients and that the HSP70 level was elevated in vascular endothelial cells in advanced diabetic patients during the inflammatory phase [55].

The ageing of blood vessels is a physiological phenomenon, and the loss of homeostasis in protein levels has been indicated in this process. This may be due to decreased expression of chaperone proteins including HSP70 [63]. HSP70 protein levels vary with age in adult and older animals, where the level of HSP expression is elevated with age. Thus, the HSP70 level may be used as a biomarker of tissue ageing [64]. HSP72 also has anti-inflammatory effects in cells that are elicited by suppression of TNF-α, interleukin (IL)-1β, and IL-6. Reduced HSP72 expression has been associated with worsening inflammation of blood vessels. Chronic inflammation and oxidative stress have been attributed to the progression of atherosclerosis. HSP70 is involved in the regulation of oxidative stress and inflammation, so HSP70 may also regulate the rate of ageing [64]. Thermal stimulation and subsequent increase in the chaperone function would elevate the level of HSP70 and delay the vascular ageing process. HSP70 is also involved in the regulation of chronic inflammation and oxidative stress in blood vessels. Thus, monitoring the degree of atherosclerosis and the HSP70 level may be useful to evaluate the preventive effect of ASCVD and to identify risk factors early in the disease process.

As mentioned above, HSP70 reduction may be a key event in the development of atherosclerosis. However, others have reported that there is a positive correlation between atherosclerosis and plasma HSP70 levels. Diet-induced atherosclerosis is associated with elevated circulating HSP70 levels in rats, suggesting that dietary release of his HSP70 (HSPA1A) may act as a paracrine factor regulating vascular homeostasis [65]. In response to various stressors, vascular cells produce high levels of HSPs to block inflammation and maintain homeostasis against stress. In contrast to reports of the chaperone function of intracellular HSP70, the role of circulating HSP70 has a more complex association with atherosclerosis [65]. Xie. F [65] reported that diet-induced elevation of eHSP70 could trigger cell adhesion with the help of IL-6 as a mediator [64]. The role of HSP70 in atherosclerosis is expected to be verified in the future. On the other hand, Zaho. ZW [60] pointed out that HSP70 promotes the progression of atherosclerosis in apoE -₋/₋ mice by suppressing the expression of ABCA1 and ABCG1 via the JNK/Elk-1 pathway.

However, the relationship between HSP70 and atherosclerosis has not yet been fully elucidated [65]. It may be possibly due to differences in model species and the complex actions of soluble HSP70 in the microenvironment [65].

Bruxel. MA [61] suggests that chronic heat treatment reverses the progression of atherosclerosis. The senescence-associated secretory phenotype observed in diabetes and obesity induces HSF1-dependent HSP expression by heat shock (HS) response, which may be suppressed by the progression of atherosclerosis [61]. Bruxel. MA [61] reported that in adult atherosclerotic mice, the expression of SIRT1, HSF1, HSP27, HSP72, and HSP73 was suppressed in the aorta in parallel with increased expression of the NFκB -dependent VCAM1 adhesion molecule. However, a group that received systemic heat treatment completely reversed the suppression of these HS -responsive proteins and markedly inhibited both VCAM1 expression and NFκB DNA-binding activity [61]. Heat treatment increased serum HDL cholesterol levels and reduced plasma levels of TG, total cholesterol, LDL cholesterol, oxidative stress, fasting glucose, insulin resistance, and mortality [61].

It has been indicated that overexpression of HSP72 is an effective intervention to control blood glucose levels and suppress chronic mild inflammation in patients with severe obesity and muscular dystrophy [40] as well as in individuals with limited physical activity levels who live in bed all day and use a wheelchair or are elderly [65]. Demyanenko. SV [33] reported that eHSP70 reduces apoptosis and necrosis in glial cells, but not in neurons, indicating that eHsp70 also has neuroprotective effects. As stated in the apoptosis section, increased eHSP70 expression is considered to have little risk of enhancing the atrophy of muscles and nerves. Aiming for a safer method of utilization, it is necessary to carry out repeated verifications.

Findings from previous studies and future ideas were summarized in Figure 2.

## 5. Conclusions

Current evidence suggests that HSP70 may be effective in preventing ASCVD by improving its underlying pathology and diseases such as T2DM and atherosclerosis. The effect of HSP70 can be enhanced by acute or chronic thermal stimulation and be elicited systemically by targeting skeletal muscle since it regulates whole-body metabolism. In addition, when using HSP70 as a target for ASCVD prevention, it is necessary to consider the ratio of iHSP70 and eHSP70 depending on the subjects with underlying pathology. We believe that HSP70-targeted interventions have the potential to regulate atherosclerosis, insulin resistance, inflammatory conditions, and other underlying pathologies of ASCVD. Although there is a consensus in previous studies that thermal stimulation is effective in improving glucose metabolism, it is important to further verify the effect of thermal stimulation on atherosclerosis and underlying chronic inflammation. It is possible that iHSP70 and eHSP70 are affected by the progress of atherosclerosis and the length of the insulin resistance period in individuals. It is necessary to verify the total life habits such as eating, exercise, and heat.

In order to prevent ASCVD, thermal stimulation might be an important method of HSP70 activation in individuals who are limited to perform physical activities because of severe obesity or locomotive syndrome. Carefully designed clinical studies are needed to ensure its safe and effective thermal application for ASCVD prevention in such subjects.

## Figures and Tables

**Figure 1 biomolecules-13-00867-f001:**
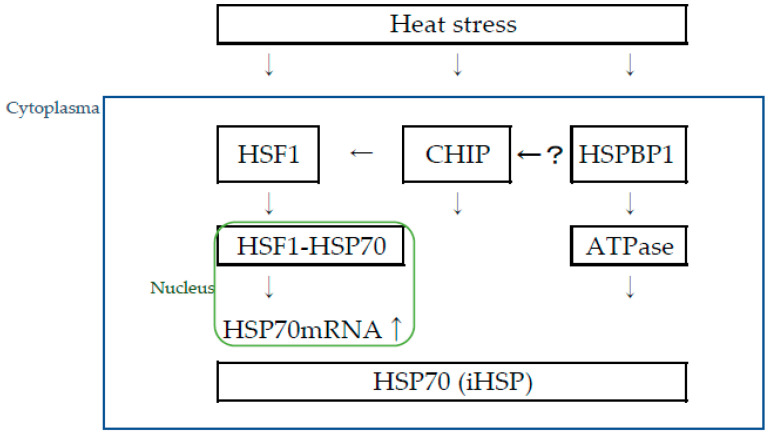
Pathway through which HSP70 is induced by thermal stimulation of skeletal muscle. HSF1: heat shock factor 1, iHSP: intracellular HSP70, eHSP70: extracellular HSP70, CHIP: carboxy terminus of Hsp70-interacting Protein, HSPBP1: heat shock protein binding protein 1, “?”: HspBP1 is an inhibitor of CHIP, but it is still unknown whether HspBP1 cooperates with CHIP in regulating the heat shock response.

**Figure 2 biomolecules-13-00867-f002:**
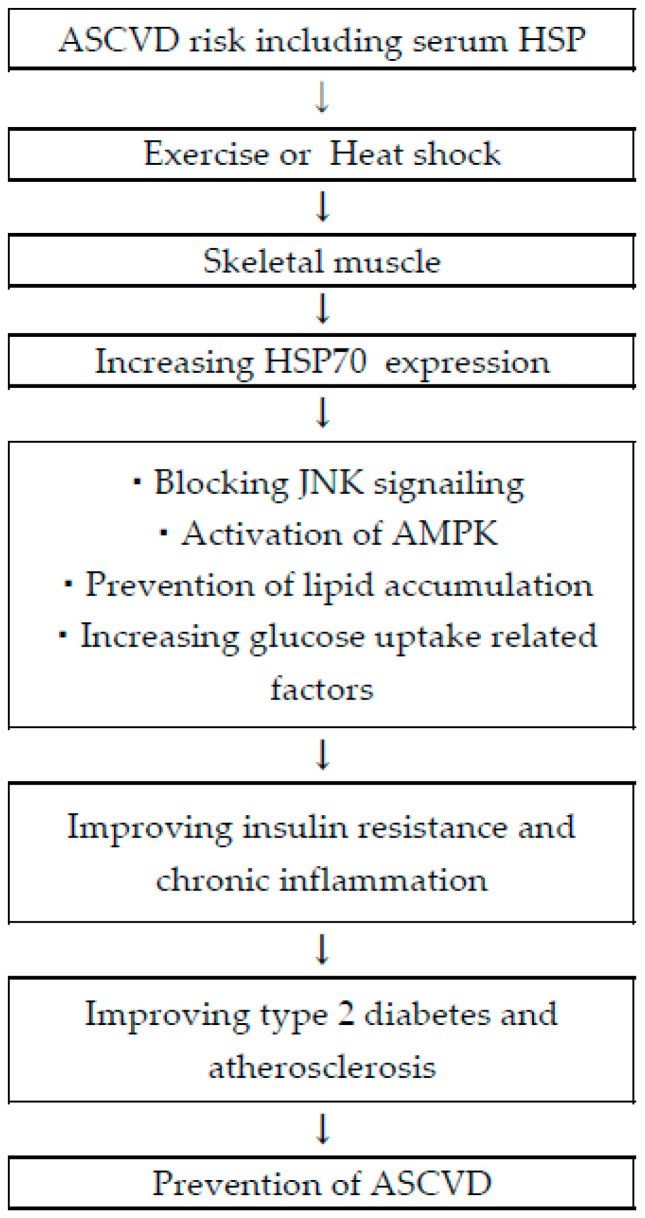
Summary of the possibility of ASCVD prevention using HSP70.

**Table 1 biomolecules-13-00867-t001:** Summary of our research results using thermal stimulation.

Methods	Results	Reference
Human SMDC cultured in vitro for 20 hours at 42 °C. mRNA levels were compared using microarray. Analysis was performed by Gene Springs.	[up regulated mRNA] HSPA1A/HSPAB, CLU, SESN3, HSP105/110, GDF15,CTRP1/C1QTNF1, SESN1, LEPR, INSR, THRA, KLF15, IGF1R, HRH1, IRS2, CEBPD [down regulatedmRNA] DPP-4, LIPG, MEF2C, HSPA, IGFBP3, ADAM12, ATF6, IGFBP1, HB-EGI	[23]
Human in vivo study using thermal sheets for ten weeks (8 hours per day, 5 days per week) Serum adiponectin and suPAR levels were measuresd by ELISA	Increased serum adiponectin and decreased serum suPAR	[24]
Human in vivo study using 4 types of hot packs (hot towels, hot packs, red bean bags, and thermal sheets) for 10 minuts -5 hours. Serum HSP70 levels were measured by ELISA.	Serum human HSP70 levels were not increased by 4 types of hot packs (hot towels, hot packs, red bean bags, and thermal sheets) Serum human HSP70 concentration was decreased with hot towel	[16]

**Table 2 biomolecules-13-00867-t002:** HSP70 and pathologic changes leading to cardiometabolic diseases.

Category	Finding	Reference
Protection of insulin resistance	Higher levels of skeletal muscle HSP70 protect against insulin resistance development during healthy aging.	[27]
	Heat treatment protects skeletal muscle from high-fat diet-induced insulin resistance. HSP induction in skeletal muscle could be a therapeutic approach to obesity-induced insulin resistance.	[28]
	Acute heat treatment can increase insulin-stimulated glucose uptake in aged skeletal muscle, with the underlying mechanism likely to be HSP72-mediated JNK inhibition.	[29]
	Intestinal secreted HSPs are one of the causes of insulin resistance when undergoing duodenal jejunal bypass.	[30]
Anti-inflammation	iHSP70 exerts potent anti-inflammatory effects and is associated with anti-insulin resistance in skeletal muscle. Increased eHSP70 is associated with inflammatory and oxidative stress conditions. An imbalance in the eHSP70/iHSP70 ratio is a contributing factor to the chronic inflammatory state that leads to the development of insulin resistance and type 2 diabetes.	[4]
Anti-apoptosis	iHSP70 inhibits apoptosis by inhibiting nuclear factor kappa B activation, nitric oxide production and superoxide dismutase activity. eHSP70 is released extracellularly under stress conditions and becomes an inflammatory mediator. Proposed eHSP70/iHSP70 as markers of inflammatory status.	[17]
	Inhibits stress-induced apoptotic signals.HSP70 synthesis act as an essential recovery mode for cellular survival and adaptation during lethal conditions.	[31]
	Sustained JNK activation promotes apoptosis.	[32]
	eHsp70 reduces apoptosis and necrosis of glial cells, but not neurons.eHsp70 reduces levels of the p53 protein apoptosis promoter (neuroprotective effect).	[33]
	Intranasal injection of recombinant Hsp70 reduced the level of apoptosis in the ischemic penumbra and stimulated axon formation.Exogenous Hsp70 significantly reduced light-induced apoptosis and necrosis of glial cells. Human recombinant Hsp70 may be translatable into clinical therapy.	[34]
Suppression of muscle atrophy	Exposure to heat stress increases HSP70 and suppresses muscle atrophy.	[35]
	Hyperthermia significantly decreases muscle atrophy markers (CD68, KLF, and MAFbx) and significantly increases muscle hypertrophy markers (AKT, mTOR, and HSP70).Hyperthermia is a viable treatment for reducing muscle wasting in diabetic patients.	[36]
Protein repair	Bathing in chloride hot springs enhances immunity such as NK cell activity, enhances protein repair function by HSP70, and has a psychological relaxing effect immediately after bathing.	[37]
	Serum-sensitive HSP70 is affected by exercise recovery, damaged tissue repair processes, and reduction of oxidative stress by inflammation. HSP70/90 expression is a potential biomarker for the preventive effects of exercise for the treatment of various diseases.	[38]

**Table 3 biomolecules-13-00867-t003:** HSP70 family and markers as well as management of cardiometabolic diseases.

Category	Finding	Reference
Diabetes	Decreased concentration of HSP70 is able to induce inflammation process through JNK activation, inhibit fatty acid oxidation by mitochondria through mitophagy decrease and mitochondrial biogenesis, as well as activate SREBP-1c, one of the lipogenic gene transcription factors in ER stress. long-term physical exercise, hot tub therapy, and administration of alfalfa-derived HSP70 in subjects with insulin resistance are proven to have therapeutic and preventive potency that are promising in T2DM management.	[25]
	There is a negative correlation between HSP70 concentrations and insulin levels and HOMA-IR, which is associated with increased activity of antioxidant enzymes.Measurement of eHSP70 concentration can be an important indicator in impaired glucose homeostasis.	[52]
	Serum levels of HSP70 are significantly higher in diabetic patients and correlate with disease duration. High HSP70 levels in long-term diabetes may be an indicator of metabolic disturbances in the course of diabetes.	[55]
	The presence of diabetes in patients with endometrioid cancer results in an increase in eHSP70.	[56]
	HSP72 contributes to the protection of β-cells against apoptosis and plays an important role in maintaining the increased demand for insulin secretion due to insulin resistance. Targeted induction of her HSP72 in the liver could be one of the therapeutic strategies for insulin resistance and T2D.	[41]
	Skeletal muscle iHSP72 and HSF-1 protein levels are reduced in type 2 diabetes. iHSP72 protein expression is associated with obesity levels and may be involved in pro-inflammatory conditions. Plasma eHSP72 is increased in obese diabetic patients. Obesity and its complications are the main cause of her elevated eHSP72, as eHSP72 is attributed to protein damage and oxidative stress levels that occur over time in the disease. eHSP72 is a potential new biomarker in diabetes.	[7]
Obesity	Obesity-induced inflammation promotes insulin resistance, impairs insulin signaling and reduces HSP expression, making tissues more susceptible to damage. The resulting damage to pancreatic beta cells leads to further loss of insulin signaling and decreased anti-inflammatory HSPs.Obesity and a sedentary lifestyle perpetuate this cycle, but diet and exercise raise her HSPs, reduce inflammation, and improve insulin signaling.	[57]
NAFLD	Heat Therapy may improve systemic metabolism through induction of hepatic HSP72. Therapies targeting her HSP72 in the liver may prevent NAFLD	[19]
Atherosclerosis	Overexpression of HSP27 is protective against atherosclerosis, whereas overexpression of HSP60 is atherogenic. HSP70 is overexpressed in advanced lesions of atherosclerotic plaques and may have a protective effect of HSP70 stimulation. The effect of HSP70 on atherosclerosis is under discussion.	[7]
	Plasma Hsp70 levels are associated with the risk of acute coronary syndrome.	[58]
	HSP70 ameliorates oxidative stress after myocardial infarction injury. Increased HSP70 levels mitigate damage from inhibition of NF-κB activity in myocardium after ischemia/reperfusion injury.	[59]
	HSP70 increases arterial lipid accumulation and promotes atherosclerotic lesion formation.	[60]
	Chronic whole-body heat treatment is involved in the anti-inflammatory and anti-aging SIRT1-HSF1-HSP and attenuates the development of atherosclerosis.	[61]

## Data Availability

Not applicable.

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
