# Peer review of "Thermal Effect on Heat Shock Protein 70 Family to Prevent Atherosclerotic Cardiovascular Disease"

_biomolecules, 2023, doi:10.3390/biom13050867_

Round 1

Reviewer 1 Report

In my opinion the work is not a significant contribution to the field, even if it is well written. The conclusion are short and the critical point of view is missing. Please modify the text accordingly. For the benefit of clarity the authors should add a table in wich are reported the principal findings of the references cited.

Author Response

Thank you for your comments on our manuscript.

We have changed the configuration throughout. I've made all the corrections, but I've put markers in the word file where I've made major changes. Regarding changes to the content, I added more citations from literature within the past five years, added critical perspectives to each item, and added them after comprehensive consideration.

  We have also tabulated the results in terms of how HSP70 is related to ASCVD in the cited literature. We also summarize our results in a table.

Reviewer 2 Report

This review focuses on the role of thermal effects on HSP70 of blood serum and skeletal muscles in cardiovascular pathologies. Members of the HSP70 family are also known to function as chaperones and are involved in a number of physiological processes, and they have recently attracted attention as a target for disease prevention and a possible biomarker for a number of diseases, which determines the relevance of this work.

Comments and Recommendations.

The literature should be expanded and updated, many sources date back to 2000-2005. For example:
Rodríguez-Iturbe B, Johnson RJ. Heat shock proteins and cardiovascular disease. Physiol Int. 2018;105(1):19-37.doi:10.1556/2060.105.2018.1.4
"The HSP70 family also functions as a chaperone and is involved in a number of physiological mechanisms. " - provide a link to contemporary sources.
"Thus, it has recently gained attention as a target for disease prevention and a a possible biomarker of disease." - give a link to modern sources.
"Their expression is induced by various stressors, including oxidative stress [1], heat stress [2], ischemia [3], exercise [1], and metabolic stress [4]." - add references to contemporary sources.

Some introductory information to update the topic of this review: about cardiovascular vascular diseases and varieties of atherosclerotic cardiovascular diseases, their dissemination. Specify their main causes and pathogenesis: atherosclerosis, insulin resistance, inflammation, cell death, etc., which will be further discussed in chapters.

Add a separate section where members of the HSP70, distribution of the protein in tissues and intracellularly. Features of HSP70 in skeletal muscles. For example, it is an interesting fact that three members of the Hsp70 family, whose appearance in the cell is induced by stress, differently target intracellular structures after thermic exposure, in particular on nucleolar structures. Three factors are known to be involved in processes which regulate the activity of Hsp70: heat shock transcription factor 1 (HSF1), a regulator of HSP gene expression; CHIP (C-terminus of Hsp70-binding protein, Hsp70 co-chaperone), capable of enhancing Hsp70 expression by activation of the previously mentioned HSF1 [
https://pubmed.ncbi.nlm.nih.gov/16554822/]; and Hsp70-binding protein 1 (HspBP1), also co-chaperone of Hsp70, which is capable of inhibiting the activity of Hsp70-ATPase [https://pubmed.ncbi.nlm.nih.gov/25487014/], and prevent CHIP-mediated ubiquitination on substrates to be cleavaged by the proteasome [https://pubmed.ncbi.nlm.nih.gov/15115282/]. The authors should consider all of these factors in the context of content levels, intracellular distribution, and expression of HSP70 in skeletal muscles in normal conditions and in cardiovascular pathologies.
Include a table summarizing the literature data and demonstrating the effects of HSP70 upon thermal effects in skeletal muscles and blood serum in atherosclerotic cardiovascular vascular disease (on which models, what effects, and references)

The review has a narrow focus: it focuses on the involvement of a particular HSP70 protein in atherosclerotic cardiovascular disease. Therefore, readers would expect to see specific mechanisms of its involvement in the form of signaling pathways, represented as a figure. Therefore, the review should be supplemented with figures that summarize and illustrate each chapter of the review. Without illustrative material, the text is difficult to comprehend. Where authors refer to their own research, it is recommended to provide formatted figures with their own results. Where the authors discuss serum HSP70 levels, it is worth mentioning methods of identification, their classification, specificity, and specifics of detection. The effects of HSP70 in thermal exposure have not been sufficiently considered. For example, the participation of the p53 protein in the implementation of anti-apoptotic effect of HSP70 in norm and under thermal exposure or other stress [
https://pubmed.ncbi.nlm.nih.gov/35011655/;
https://pubmed.ncbi.nlm.nih.gov/32870479/].

Throughout the review, the authors refer to endogenous HSP70. What effects can be achieved by the administration of an exogenous protein in the context of these pathologies? Are there such studies available? This would embellish and emphasize the
the practical aspect of the review.

Author Response

Thank you for your comments on our manuscript.

References have been updated to add or correct content for each item.

We added a new section on HSP70 activity, explaining that HSP70 is associated with underlying ASCVD. This is also added to each section. In addition to adding a figure on HSP70 activity, the results of the cited literature and the results of our previous research are also shown in the table. We added separate descriptions of iHSP70 and eHSP70 in each section. I'm making changes to the overall configuration and fixing the content.

I am attaching the revised manuscript and figure files.
Thank you for your kind support.

Reviewer 3 Report

Authors review evidence suggesting that HSP70 may be effective in preventing atherosclerotic cardiovascular disease (CVD) by improving pathology and diseases associated with it or involved in its pathogenesis. They conclude that that the effect of HSP70 can be enhanced by acute or chronic thermal stimulation and be elicited systemically by targeting skeletal muscle since it regulates whole-body metabolism. Authors propose to develop thermal stimulation-based preventive modality for individuals who are limited in performing physical activities.

I support the idea proposed in the review in general as it is a promising strategy for developing non-pharmacological method to prevent atherosclerotic CVD in individuals who cannot do exercise for CVD prevention. The explanation of the mechanistic basis of new prevention strategy at molecular level seems relevant. However, this manuscript cannot be published in its current form without improving it in the following aspects:

It is challenging to assess the rigor of a number of references (over a dozen) as they are published in journals not indexed by PubMed, Scopus, or Web of Science, which guarantee quality of peer-review in the indexed journals. These References are as follows: 6, 9, 10, 11, 14, 19, 27, 35, 37, 38, and 39 as well as Ref. of papers published in the Journal of Advances in Medicine and Medical Research (Ref. 29, and Ref. between 5 and 6, 7 and 8).

Authors state in the conclusion paragraph that “Current evidence suggests that HSP70 may be effective in preventing ASCVD by improving its underlying pathology and diseases such as type 2 diabetes and atherosclerosis”. However, the reviewed evidence is not that “current” as the mean year of publication of cited sources is 2009 ± 10 years. Only 12 references have been published within the last five years; it is less than 18% of the entire pool of reviewed literature. Proportion of literature sources published within the last five years should be increased (ideally up to 50%).

Reference list is formatted inconsistently. Order of references is sometimes messed up (between Ref. 5 and 6, 7 and 8, 48 and 49). Style of reference formatting is also inconsistent. Some links are misleading and should be updated.

Consistency of referencing literature sources must be improved.

Quality of figure should be increased as it has low resolution and type fonts that seem somewhat distorted.

Authors should discuss other mechanisms, which are up- or downregulated by thermal shock. It is essential considering that thermal treatments, even topical, usually are discouraged and should be avoided in individuals with predisposition to and with manifested rheumatoid arthritis, other autoimmune disorders, and some benign tumors, which potentially can turn to malignant. It would be helpful to discuss strategies to avoid the potential side effects of thermal treatments such as triggering autoimmune response in predisposed individuals especially in those with locomotive syndrome due to rheumatoid arthritis.

I would encourage authors to provide the section on balneology with more detailed discussion of therapeutic potential of hot springs.

Author Response

Thank you for your comments on our manuscript.

References have been updated to add or correct content for each item. We have also modified the format of the list of references. We fixed the resolution of the figure.

As for the contents, the composition has been changed as a whole. We added a new section on HSP70 activity, explaining that HSP70 is associated with underlying ASCVD. This is also added to each section. In addition to adding a figure on HSP70 activity, the results of the cited literature and the results of our previous research are also shown in the table. Added separate descriptions of iHSP70 and eHSP70 in each section.

He also added that there is no consensus on the effects of HSP70 on muscle atrophy and rheumatoid arthritis. Also added a description of locomotive syndrome and balneotherapy.

I am attaching the revised manuscript and figure files.
Thank you for your kind support.

Round 2

Reviewer 2 Report

The authors have done a nice job of revising this manuscript and addressing the reviewer comments and queries. The manuscript now reads with greater focus.

Reviewer 3 Report

Dear Authors, thank you for your revisions.